# Video triage of children with respiratory symptoms at a medical helpline is safe and feasible–a prospective quality improvement study

**Caroline Gren** [1,2]*, **Asbjoern Boerch Hasselager**[3], **Gitte Linderoth**[2,4,5], **Marianne Sjølin Frederiksen**[6], **Fredrik Folke**[2,5,7], **Annette Kjær Ersbøll** [5,8], **Hejdi Gamst-Jensen**[9,10], **Dina Cortes**[1,2]

1 Department of Pediatrics and Adolescence Medicine, Copenhagen University Hospital–Amager and Hvidovre, Copenhagen, Denmark, 2 Department of Clinical Medicine, Faculty of Health and Medical Sciences, University of Copenhagen, Copenhagen, Denmark, 3 Department of Pediatrics and Adolescence Medicine, Copenhagen University Hospital–Nordsjællands Hospital, Hillerød, Denmark, 4 Department of Anesthesia and Intensive Care, Copenhagen University Hospital–Bispebjerg and Frederiksberg, Copenhagen, Denmark, 5 Copenhagen University Hospital–Copenhagen Emergency Medical Services, Copenhagen, Denmark, 6 Department of Pediatrics and Adolescence Medicine, Copenhagen University Hospital–Herlev and Gentofte, Copenhagen, Denmark, 7 Department of Cardiology, Copenhagen University Hospital–Herlev and Gentofte, Copenhagen, Denmark, 8 National Institute of Public Health, University of Southern Denmark, Odense, Denmark, 9 Department of Clinical Research, Copenhagen University Hospital–Amager and Hvidovre, Copenhagen, Denmark, 10 Department of Emergency Medicine, Copenhagen University Hospital–Amager and Hvidovre, Copenhagen, Denmark

* ida.caroline.gren.02@regionh.dk

## Abstract

### Background

Young children are among the most frequent patients at medical call centers, even though they are rarely severely ill. Respiratory tract symptoms are among the most prevalent reasons for contact in pediatric calls. Triage of children without visual cues and through second-hand information is perceived as difficult, with risks of over- and under-triage.

### Objective

To study the safety and feasibility of introducing video triage of young children with respiratory symptoms at the medical helpline 1813 (MH1813) in Copenhagen, Denmark, as well as impact on patient outcome.

### Methods

Prospective quality improvement study including 617 patients enrolled to video or standard telephone triage (1:1) from February 2019-March 2020. Data originated from MH1813 patient records, survey responses, and hospital charts. Primary outcome was difference in patients staying at home eight hours after the call. Secondary outcomes weas hospital outcome, feasibility and acceptability. Adverse events (intensive care unit admittance, lasting

**Data Availability Statement:** The present study was conducted as a quality improvement project and was approved by the local hospital

managements and the Emergency Medical Services. The study was therefore not registered with the regional and/or national research boards, and written consent were not obtained from the participants. The study was furthermore assessed by the Research Ethics Committee in the Capital Region of Denmark, who deemed that approval was not indicated. As such, sharing of sensitive patient data is prohibited by Danish legislation and international General Data Protection Regulations (GDPR). With referral to the Acceptable Data Access Restrictions of PLOS, we have prepared an anonymized minimal data set, which can be accessed by reasonable request to nanna.birch. andersen@regionh.dk at Copenhagen Emergency Services.

**Funding:** The study was funded by the Danish foundation TrygFonden (ID 124362; awarded to ABH; www.tryghed.dk), the Research Foundation at Amager Hvidovre Hospital (no ID; awarded to CG; https://www.hvidovrehospital.dk/forskning/Sider/default.aspx) and the Research Foundation of the Capital Region (A6207; awarded to DC; https://www.regionh.dk/english/research-and-innovation/Pages/default.aspx). The funders had no role in study design, data collection and analysis, decision to publish, or the preparation of the manuscript.

**Competing interests:** The authors have declared that no competing interests exist.

injuries, death) were registered. Logistic regression was used to test the effect on outcomes. The COVID-19 pandemic shut the study down prematurely.

## Results

In total, 54% of the included patients were video-triaged., and 63% of video triaged patients and 58% of telephone triaged patients were triaged to stay at home, (p = 0.19). Within eight and 24 hours, there was a tendency of fewer video-triaged patients being assessed at hospitals: 39% versus 46% (p = 0.07) and 41% versus 49% (p = 0.07), respectively. At 24 hours after the call, 2.8% of the patients were hospitalized for at least 12 hours. Video triage was highly feasible and acceptable (>90%) and no adverse events were registered.

## Conclusion

Video triage of young children with respiratory symptoms at a medical call center was safe and feasible. Only about 3% of all children needed hospitalization for at least 12 hours. Video triage may optimize hospital referrals and increase health care accessibility.

## Introduction

Contacts to out-of-hours (OOH) healthcare services are increasing globally and contribute to emergency department (ED) crowding, which has a negative impact on patient safety [1, 2]. Call centers are used to optimize the use of acute OOH services by gatekeeping and triage of callers instead of self-referrals, and may further support the callers by providing advice on self-care [3]. Telephone triage is safe [4, 5], both when performed by doctors and nurses [6].

Contacts regarding young children is a frequent cause for contacts at call centers and in acute care settings [7, 8]. Even though these children often have mild symptoms that do not require urgent treatment [9], help-seeking is often catalyzed by parental worry and a feeling of lack of control [10–13]. The contact is often due to a feeling of responsibility of "doing everything" in order to care for the child [10, 14, 15], the fear of doing something wrong [16], and to rule out a serious disease [12, 17].

Telephone triage lacks visual cues, and information obtained through secondhand information is perceived as challenging by call-handlers, which is the case when parents call on behalf of their children [18, 19]. This challenge might interfere with the overarching goal of telephone triage to identify and meet the caller's needs (20), in a patient-centered manner [21]. Live video may facilitate the assessment and communication, as the call-handler can observe the child directly. Most children seen at pediatric EDs have symptoms from the respiratory tract [22, 23]. Telemedicine is reliable when assessing respiratory symptoms in children at pediatric EDs according to two studies investigating the correlation between face-to-face and telemedicine observations[24, 25]. Another study reported that video transmission of children with acute asthma from parents' smartphones in the hospital waiting area was feasible and that both doctors and parents found it reassuring and useful [26].

This study investigated video streaming of children with symptoms from the respiratory tract as part of the primary triage at a medical call center, in comparison with standard telephone triage, and studies whether video triage in such cases is safe, feasible, and acceptable. Our primary aim was to investigate whether video triage could result in a significant increase in parents who could stay home with self-care of their child, without under-triaging acutely

sick children; in other words, that it was safe. This paper only reports patient outcome, and we have described the experiences of the parents and the call-handlers in a separate paper [27].

## Methods

This was a prospective quality improvement study that investigated the introduction of video triage at a medical call center. The study has been reported in accordance with the Standards for Quality Improvement Reporting Excellence (SQUIRE) 2.0 guidelines for quality improvement studies [28].

### Context

This prospective quality improvement study was conducted at the medical helpline 1813 (MH1813), a helpline for OOH calls regarding acute, non-urgent medical illness and injury. MH1813 is part of the Emergency Medical Services (EMS) Copenhagen, Denmark and receives approximately 1 million calls per year from a catchment population of 1.8 million inhabitants. Approximately 25% of the calls concern children younger than 12 years of age [29]. Citizens are encouraged to call MH1813 when they experience acute health issues OOH, and telephone contact is mandatory before seeking face-to-face consultation at hospitals. Most calls are handled by registered nurses, and the rest by physicians, either primarily or after a nurse has forwarded a call. In general, healthcare services in Denmark are paid by taxes, including visits to general practitioners (GP), hospitals, and the use of EMS.

When a citizen calls MH1813, a call-handler assesses the caller's need for medical help and chooses an appropriate response. Nurses must use an electronic triage tool, which guides them regarding what questions to ask and the subsequent appropriate response. In calls regarding children, the possible outcomes are: (1): the child stays at home (self-care and/or contact to GP the next workday if needed) or (2) the child is referred to hospital (pediatric urgent care clinic or pediatric ED). In the pediatric urgent care clinics, prescriptions can be made, but no on-site treatment can be provided and only a few point-of-care tests can be performed. In suspected life-threatening conditions, the call is forwarded to the EMS ambulance service.

### Intervention

Video streaming was carried out using GoodSAM Instant-on-Scene function™ technology (https://www.goodsamapp.org/). It is a browser-based streaming technology, which means that no installation of an application at the parents' smartphone was necessary. The call-handler sent a text message to the parent, who, upon giving consent to sharing location and video, streamed a live video feed to the call-handler, Fig 1. The technology is described in detail elsewhere [30].

After all calls, the call-handlers answered an electronic survey about their experiences and sent an electronic survey to the parents. Workflows are shown in S1 Appendix.

All participating call-handlers were nurses. They were selected by the management as suitable participants as they were experienced call-handlers with technical aptitude. The call-handlers received a theoretical and practical introduction to the project by project coordinator (CG), who also was available in the call center for technical support and assistance regarding project workflow. No support was provided regarding the triage itself. Initially, 20 call-handlers participated. More were continuously incorporated if call-handlers withdrew.

**Inclusion and exclusion criteria.** All children aged six months to five years presenting with respiratory symptoms were eligible for inclusion. We did not include children younger than six months as there are higher risks of them having severe illness, and they are inherently more difficult to assess. Also, this age span includes the children that most often present to

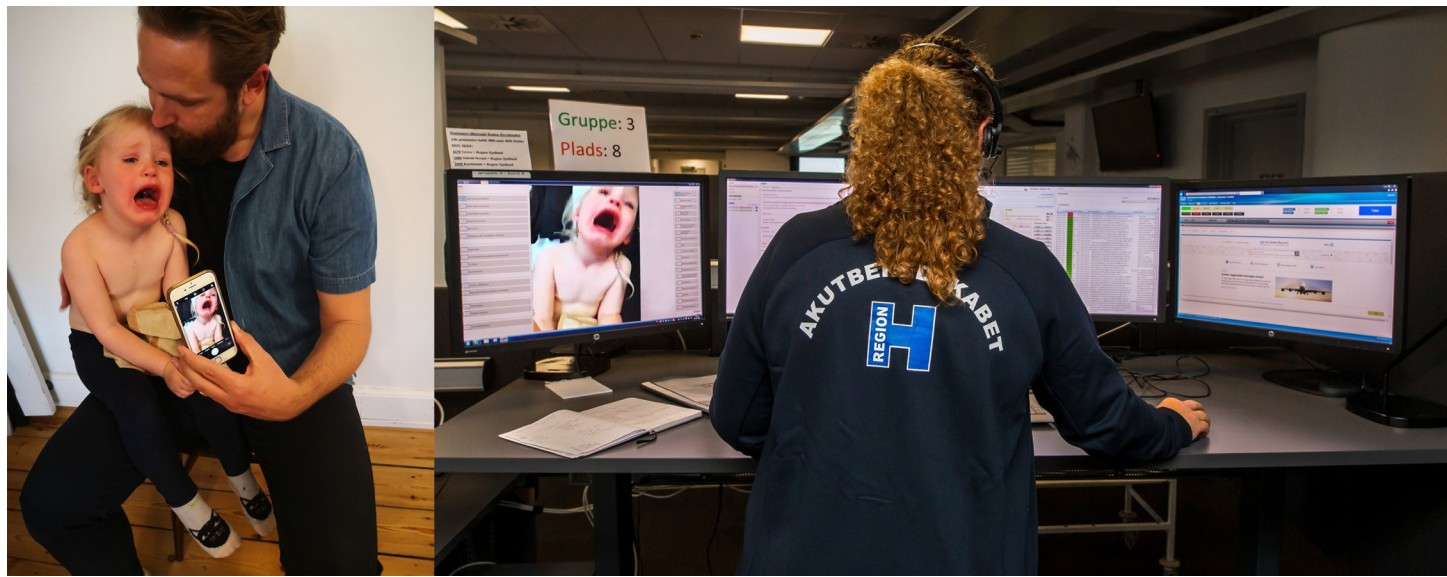

**Fig 1. Parent streaming video to call-handler at MH1813.** MH1813: Medical Helpline 1813.

MH1813. Video triage might be extra beneficial in this age group, as these children not always are able to verbally explain their symptoms due to their developmental stage. Participation was voluntary and parents were informed that they would receive standard care (telephone assessment) if they declined to participate in video triage. Parents in the telephone group were informed about the project and asked if they agreed to receive a survey by text message.

Exclusion criteria were foreign telephone number, previous participation in the project, neither Danish- nor English-speaking parent, unsuccessful video streaming, not calling from a smartphone, and calls without information about triage method; i.e. telephone or video triage.

**Sample size.** In 2018, 177,000 calls were made to MH1813 regarding children up to and including five years, excluding injuries [29]. Of these calls, 56% were advised to stay at home with self-care and 42% were referred to hospitals (the remaining 2% were referred to outpatient clinics or received prescription). We expected that video triage would increase the proportion of parents staying at home providing self-care for the child with 10 percentage points when they called MH1813, i.e. an increase from 56% to 66%. With a power of 80% and 5% significance and a two-sided test, 774 children had to be included, divided into two equal groups, to identify such a difference.

**Patient inclusion.** The call-handlers included patients to the video triage group or to the telephone triage group at a 1:1 ratio. The call-handlers were instructed to include *all* eligible patients and to maintain allocation to the video triage group one workday, and to the telephone triage group the next workday, and so on.

However, if the waiting time for the callers was too long, inclusion in the project was temporarily paused, either at the discretion of call-handlers or by the floor manager.

After the first two weeks, the pilot results revealed that no adverse events had occurred, that there was a high level of satisfaction from parents and call-handlers, and that the technical set-up of video streaming was easy to use. Later, the allocation mode was changed to every other call–i.e. one call was video triaged, the next was telephone triaged, and so on–in order to more easily remember the allocated triage group.

In March 2020, the COVID pandemic struck Denmark and caused an extraordinary workload at MH1813. Therefore, the project was stopped when a total of 617 patients were eligible for analysis, including the pilot period.

**Outcomes.** The primary outcome was the difference in the number of patients staying at home for eight hours after the MH1813 call (video triage vs. telephone triage). Secondary outcomes were patients assessed at hospitals with or without subsequent hospitalization within eight, 24, and 48 hours after the call, and the clinical outcome of patients examined at hospitals. Moreover, we studied whether video triage was safe, feasible, and acceptable. Safety was defined as meaning that no seriously sick children in the video triage group were referred to stay at home, resulting in a delayed treatment, and also by the occurrence of adverse events, admittance at intensive care units (ICU), lasting injuries or death. Feasibility was defined as how many video calls were successfully conducted after initiation and acceptability as the number of parents agreeing to participate.

Parents' and call-handlers' experiences with video triage were reported in a separate paper [27].

## Measures

Project data used for this paper were derived from three sources: the patient records at MH1813, survey responses of call-handlers, and the hospitals' patient charts.

**The patients.** Data on included patients were extracted from the electronic patient records at MH1813, survey responses, and hospitals' patient charts. Follow-up was done within 2–8 workdays after the call. We registered whether the patients were seen in a face-to-face consultation within eight, 24, or 48 hours after the call at one of the region's hospitals. Information on time, location and International Statistical Classification of Diseases and Related Health Problems (ICD-10) diagnosis was recorded, along with treatment and duration of the child's stay at the hospital. Adverse events were extracted from the hospitals' patient charts and defined as transfer to ICU, lasting injuries or death. Furthermore, the duration of video calls was compared to standard pediatric calls.

## Analyses

**Statistical analyses.** Patients' characteristics were described with frequency (number, percentage), median, and interquartile range ($Q_1$-$Q_3$). Differences between video and telephone triage groups were analyzed using chi square-test or multinomial logistic regression when appropriate, and age distribution by Mann-Whitney U-test. Multinomial logistic regression was used in the analysis of patient outcomes. An odds ratio (OR) was estimated for the multinomial logistic regression analyses with corresponding 95% confidence intervals (95% CI). A 95% CI for proportions was calculated using the Wilson Binominal Proportion Confidence Interval. P-values less than 0.05 in two-sided tests were considered statistically significant. The statistical analyses were made with SAS Enterprise Guide 7.1 (SAS Institute, Cary, NC, USA), and Open Source Epidemiologic Statistics for Public Health (www.OpenEpi.com).

## Ethical considerations and registration

The Research Ethics Committee in the Capital Region of Denmark deemed that approval was not indicated (Journal number H-18049733), and therefore participant consent was not needed. However, all participating parents were informed about the study and provided verbal consent. As the study was a quality improvement study, the managements of the hospitals in the Capital Region with pediatric departments or pediatric urgent care clinics approved access

to the patient records, as well as the management of EMS Copenhagen. The study was registered at ClinicalTrials.gov (Id: NCT03874520).

## Results

### Study population

The study population consisted of 617 children: 336 (54%) in the video triage and 281 (46%) in the telephone triage group, Fig 2. The study period lasted from February 6, 2019 to March 14, 2020.

Of the 734 parents who were initially approached, fewer than 10% (n = 35) declined to participate, and in 35 of the calls in the video triage group, the video streaming did not succeed, representing the rates of acceptability and feasibility, respectively. The distribution of gender, age and registered symptoms was almost identical in the two triage groups, Table 1. Sixty-three percent of patients in the video triage group and 58% in the telephone triage group were triaged to stay at home (p = 0.19).

### Patient outcome

The following section concerns all patients assessed at hospital within 48 hours of the study call, irrespective of the triage outcome at the end of the study call. This means that the parent may have called MH1813 again if need arose, or the child may have been referred (by a GP, for example) at a later time within 48 hours.

Within eight hours after the call, there was a tendency for fewer video-triaged than telephone-triaged patients to be assessed at hospitals (39% versus 46% p = 0.07), and also within 24 hours after the call (41% versus 49% p = 0.07). Within 48 hours there was no difference (46% versus 51% p = 0.24). For the children assessed at hospitals, the number of patients who received treatment, testing, prescriptions, or were hospitalized at least 12 hours was equal in the two groups, Table 2.

A higher number of patients were examined at a hospital after the call than the number initially triaged to hospital; in total 298 patients were assessed at hospitals within 48 hours, versus

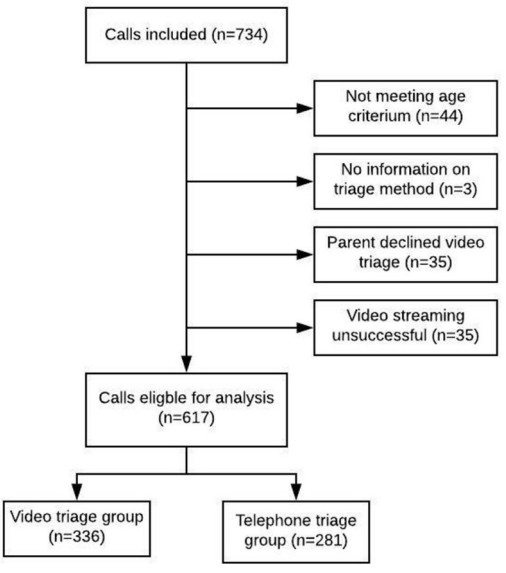

**Fig 2. Patient flow.**

**Table 1. Characteristics of patients included in video or telephone triage groups.**

| | Video triage (n = 336) | Telephone triage (n = 281) | p-value | OR (95% CI) |
|---|---|---|---|---|
| **Age, year** *median (Q₁-Q₃)* | 1.4 (1.0–2.3) | 1.5 (1.0–2.5) | 0.37* | |
| **Gender** | | | | |
| Male§ | 54% (180) | 57% (161) | 0.35** | |
| **Airway symptom registered by call-handler**§ | n = 315 | n = 244 | 0.17# | |
| Cough/difficulty breathing | 68% (215) | 65% (159) | | 1. (ref) |
| Cold | 17% (55) | 18% (43) | | 0.95 (0.60–1.48) |
| Suspected influenza | 6% (20) | 4% (10) | | 1.48 (0.67–3.25) |
| Other | 8% (24) | 13% (31) | | 0.57 (0.32–1.01) |
| Missing | 1% (1) | 1% (1) | | - |
| **Response**§ | | | | |
| Staying at home | 63% (213) | 58% (163) | 0.19** | |
| Referred to hospital | 37% (123) | 42% (118) | | |

OR: odds ratio; Q₁₋₃: Interquartile Range 1–3;

§ percentage (number);

\* Mann-Whitney U-test;

\*\* Chi-square test;

#Multinomial logistic regression; ref: reference

241 patients that were initially referred to hospital. That is, 57 patients had been referred to hospitals at a later occasion, with no significant differences between groups. Additionally, when looking at the most seriously ill patients–that is, those hospitalized for at least 12 hours–14 children were hospitalized at eight hours after the call (nine in the video group, five in the telephone group); within 24 hours another three were admitted for at least 12 hours (two in the video group, one in the telephone group); and three more after 48 hours (two in the video group, one in the telephone group). The corresponding frequencies of children staying at the hospital for at least 12 hours were 2.3% (14/617), 2.8% (17/617), and 3.2% (20/617).

Approximately 75% of the patients examined at hospitals received an ICD-10 diagnosis regarding symptoms from the respiratory tract. The unspecific diagnosis "viral infection" was most frequent in the telephone triage group, (13% vs 6%), but otherwise no differences appeared, Table 2.

A sensitivity analysis revealed no differences with or without the inclusion of patients from the pilot period; S2 Appendix. Finally, the call-handling duration of a video call in the study, including filling out and sending surveys, etc., took 3–5 minutes more than a standard pediatric call to MH1813.

## Safety

No adverse events (0/617) (95% CI 0–0.6%), admittance to ICU, lasting injuries, or deaths were registered in either group.

## Discussion

This study found that video triage of children aged six months to five years with respiratory symptoms such as cough, cold, and difficulty breathing at a medical call center is safe, feasible, and acceptable. This knowledge is valuable to other call centers that may be considering implementing video triage, as little has been published on this topic before. The primary aim of significantly increasing the number of children able to stay at home was not met, as we found a

**Table 2. Outcome of patients from the video and telephone triage groups assessed at hospital within 48 hours after the index call.**

| | Video triage 54% (336) | Telephone triage 46% (281) | p-value | OR (95% CI) |
|---|---|---|---|---|
| **Outcome at 8 hours after the call** | | | | |
| **Patients assessed at a hospital**§ | 39% (130/336) | 46% (129/281) | 0.07* | |
| **Difference in hospital outcome, video vs telephone** | | | 0.23** | |
| No treatment/paraclinical testing/prescription/hospitalization | 44% (57/130) | 46% (59/129) | | 1.0 (ref) |
| Received treatment, paraclinical testing, prescription and/or hospitalized <12 hours | 49% (64/130) | 50% (65/129) | | 1.41 (0.85–2.34) |
| Hospitalized ≥12 hours | 7% (9/130) | 4% (5/129) | | 2.15 (0.68–6.75) |
| **Outcome at 24 hours after the call**§ | | | | |
| **Patients assessed at a hospital** | 41% (139/336) | 49% (137/281) | 0.07* | |
| **Difference in hospital outcome, video vs telephone** | | | 0.15** | |
| No treatment/paraclinical testing/ prescription/hospitalization | 42% (59/139) | 46% (63/137) | | 1.0 (ref) |
| Received treatment and/or paraclinical testing and/or hospitalized <12 hours | 50% (69/139) | 50% (68/137) | | 1.48 (0.90–2.42) |
| Hospitalized ≥12 hours | 8% (11/139) | 4% (6/137) | | 2.22 (0.80–6.31) |
| **Outcome at 48 hours after the call**§ | | | | |
| **Patients assessed at a hospital** | 46% (155/336) | 51% (143/281) | 0.24* | |
| **Difference in hospital outcome, video vs telephone** | | | 0.11** | |
| No treatment/paraclinical testing/ prescription/hospitalization | 41% (64/155) | 46% (66/143) | | 1.0 (ref) |
| Received treatment and/or paraclinical testing and/or hospitalized <12 hours | 50% (78/155) | 49% (70/143) | | 1.54 (0.95–2.48) |
| Hospitalized ≥12 hours | 8% (13/155) | 5% (7/143) | | 2.13 (0.81–5.62) |
| **ICD-10 diagnoses**§ | | | | |
| **Difference in diagnoses video vs telephone** | | | 0.14** | |
| **Respiratory tract diagnoses** | 78% (121/155) | 74% (196/143) | | 1.11 (0.57–2.17) |
| **Viral infection** | 6% (10/155) | 13% (19/143) | | 2.43 (0.91–6.49) |
| **Other**¤ | 15% (24/155) | 13% (18/143) | | 1.0 (ref) |
| **All diagnoses concerning the respiratory tract** | | | | |
| *Acute upper respiratory tract infection* | 35% (55/155) | 34% (48/143) | | |
| *Acute obstructive laryngitis* | 10% (16/155) | 7% (10/143) | | |
| *Acute bronchitis* | 10% (16/155) | 12% (17/143) | | |
| *Middle ear infection* | 7% (11/155) | 8% (11/143) | | |
| *Cough WS* | 7% (11/155) | 4% (6/143) | | |
| *Pneumonia* | 4% (6/155) | 6% (8/143) | | |
| *Asthma, influenza, RSV infection, acute respiratory insufficiency* | 4% (6/155) | 4% (6/143) | | |

OR: odds ratio;

*Chi-square test;

**multinomial logistic regression;

§ percentage (number/total number); ref: reference; WS: Without specification; RSV: Respiratory syncytial virus;

¤ Other diagnoses (listed after decreasing number of patients): Fever without specification, gastroenteritis, observation due to suspected illness, acute tonsillitis, contact aborted by patient, counseling due to worry about illness in healthy person, dehydration, abdominal pain, febrile seizures, hand,—foot- and mouth disease, swelling or lump in skin or subcutis, stomatitis without specification).

non-significant difference of 5% (58% versus 63%, p = 0.19) between the triage groups. The sample size was based on an expected increase from 56% to 66%. The 56% was derived from all calls regarding children under six years of age, managed by all call-handlers in 2018. The fact that only experienced nurses were selected to participate in this study, and only a subgroup of children was included, could explain why a higher number of patients could stay at home after telephone triage (58%) in the present study and why the impact of adding video was not as large as expected. Call-handlers with less experience might gain more in terms of

reassurance and better assessment when using video streaming, as well as whether video could be used at the call-handlers' discretion. Moreover, we had to stop the study before the calculated sample size was met because of the COVID-19 pandemic. However, there is reason to believe that video triage might optimize the referral to hospitals, as there was a tendency for fewer video-triaged patients to be assessed at hospitals within both eight and 24 hours after the call (p = 0.07). The findings were non-significant, which might also be due to the study ending prematurely.

Importantly, we found that video triage was safe. No adverse events (death, ICU admittance, or lasting injuries) were registered in the study. This is in line with the results from a systematic review that studied safety in out-of-hours telephone care, which reported a safety of 97% (95% CI 96.5–97.4%) of all patients contacting out-of-hours at call centers [5]. However, that also included mortality, unplanned hospitalizations, emergency room attendance, and medical errors. Furthermore, the rate of under-referral is an important measure of how a call center performs. Previously, potential under-referral of children from a call center to pediatric ED was defined as a child who was given a nonurgent response (advised to seek care later than four hours after the call or to conduct self-care), but was hospitalized within 24 hours after the index call [4]. That study reported a potential under-referral rate of one case per 599 calls (95% CI: 1 case per 472–901 calls). However, it was stated that only about 60% of potential under-referrals may be true under-referrals, whereas the rest may reflect clinical deterioration that was not predictable at the time of the index call. In the present study, the number of patients admitted for at least 12 hours increased from 14 children at eight hours after the call to 17 children at 24 hours after the call. Therefore, the potential under-referral rate in the present study was 0.5% (3/617) within 24 hours, with no significant difference between video and telephone triage (p>0.99). In all, 57 more patients were assessed at hospital than were referred to hospital in the study call, with no difference between the groups.

Therefore, even when including the potential under-referral of three children, safety in this broader definition was 99.5% (614/617) (95% CI: 98.6–99.9%). Accordingly, video triage may help parents to safely stay at home with their sick child after parental guidance from call-handlers, as well as help identify the most appropriate course of sick children and reduce the risk of overcrowding at hospitals.

Participating in video triage in an acute and possibly stressful situation as in calls regarding sick children appears feasible and acceptable, as only 9% of video calls did not technically succeed and fewer than 10% of invited parents declined to participate in the study. However, it could be that not all parents have the capacity needed to participate in a study in the vulnerable situation of caring for a sick child. Further exploration of the users' experiences of video triage is presented in an article investigating the users' experiences [27].

We found that 5.6% of the children assessed at a hospital within eight hours after the call were hospitalized for at least 12 hours. This is somewhat lower than in previous studies, with reporting admission rates ranging from 7.6%-–10.3% [31–34]. This might reflect the fact that patients with respiratory tract symptoms are less ill than an unselected patient population, or that unknown differences exist in treatment and care strategies in Denmark than in other countries.

In adults, almost one out of four calls to an OOH helpline in another Danish region were assessed as inappropriate by the GPs answering the call; that is, the caller should have contacted their GP during daytime instead [35]. In 53% of the calls that had been classified as inappropriate by the GP, the callers assessed their problem as severe, and these calls were significantly associated with unfulfilled patients' expectations [35]. This dilemma of health professionals' and patients' perspectives and expectations not being the same might lead to discontent and possibly impeded contact and triage. Therefore, it may be of great use for

health professionals to observe children directly by video streaming, as the parents then may feel that the responsibility does not lie entirely on themselves, as well as eliminating the use of second-hand information, which is perceived as difficult by call-handlers [19].

### Limitations

We included young children with symptoms from the respiratory tract, which are the most prevalently occurring symptoms at a medical helpline and are easy to assess by video. Further research is needed to gain knowledge about whether video triage is safe and beneficial in other ages and symptoms as well.

It was not possible to conduct this study as a randomized trial because the call-handlers did not have sufficient time for thoroughly informing the parents and obtaining written informed consent. Furthermore, it was not possible to change the computer set-up to randomly allocate calls to video or telephone triage, or to change the introduction speech of MH1813 as would have been desired in terms of informing about the study. Therefore, we cannot be quite sure that the children in the video triage group are representative for the population as a whole. However, the present outcomes reflect the pragmatic, clinical use of video triage at a functioning, busy call center, and we found no differences in the number of patients receiving treatment, paraclinical testing, prescription, or hospitalization between the two groups of patients.

The study period was longer than expected and we did not reach the calculated number of included patients. This may be a consequence of longer duration of study calls than standard telephone calls, as the study calls included several additional steps compared to the normal workflow (for example, filling out and sending surveys). This led to interruption of patient inclusion during high flow periods. Incomplete inclusion of patients could also result from difficulties in implementing video triage, as it is a complex task that changes long-established routines, even though the video streaming tool was found to be easy to use. Moreover, the electronic triage tool used by the call-handlers was not changed to include video footage as part of the decision-making process. The users' experiences of video triage were described in a qualitative study, which found that the most common reason for call-handlers to opt out of including patients was a stressful work environment [27]. The parents showed little or no tendency to experience technical difficulties.

As is the case for many current studies, the inclusion of patients had to be stopped prematurely due to the COVID-19 pandemic. The findings would be more conclusive had we met the calculated sample size.

### Conclusion

Introduction of video triage was feasible and acceptable at this medical call center, and was found to be a safe method to potentially optimize hospital referrals. The number of video-triaged children who were able to stay at home was 63%, but not statistically significantly higher than 58% in the standard telephone triage group. This could be due to the number of included patients not meeting the calculated sample size as the COVID-19 pandemic shut the study down prematurely. This novel triage tool was safe and has the potential to improve accessibility to qualified assessments by health care professionals as well as using the health care workforce in a more optimal way.

### Supporting information

**S1 Appendix. Flowchart of project workflow during video- and telephone triage calls, depicting the call-handlers' workflow.**
(PDF)

**S2 Appendix. Sensitivity analysis.** Patient outcome with and without pilot group included. *Chi square test.
(PDF)

## Acknowledgments

The authors wish to thank all nurses participating in the video triage projects, they could not have been accomplished without you. At EMS Copenhagen, the authors would like to thank former Head of Emergency Medical Dispatch Center Marie Baastrup and former CEO Freddy K. Lippert, for seeing the possibilities in our study and allowing us to implement it. Moreover, thanks to Martin Vang Haugaard at EMS Copenhagen for his data support. Thank you to GoodSAM for the usage of their Instant-on-Scene™ platform, and especially Ali Ghorban-gholi for swift and competent technical support at all times.

## Author Contributions

**Conceptualization:** Caroline Gren, Asbjoern Boerch Hasselager, Marianne Sjølin Frederiksen, Fredrik Folke, Dina Cortes.

**Data curation:** Caroline Gren.

**Formal analysis:** Caroline Gren, Annette Kjær Ersbøll, Dina Cortes.

**Funding acquisition:** Caroline Gren, Asbjoern Boerch Hasselager, Dina Cortes.

**Investigation:** Caroline Gren.

**Methodology:** Caroline Gren, Asbjoern Boerch Hasselager, Gitte Linderoth, Marianne Sjølin Frederiksen, Fredrik Folke, Hejdi Gamst-Jensen, Dina Cortes.

**Project administration:** Caroline Gren.

**Resources:** Gitte Linderoth.

**Supervision:** Marianne Sjølin Frederiksen, Fredrik Folke, Hejdi Gamst-Jensen, Dina Cortes.

**Visualization:** Caroline Gren, Hejdi Gamst-Jensen, Dina Cortes.

**Writing – original draft:** Caroline Gren, Dina Cortes.

**Writing – review & editing:** Asbjoern Boerch Hasselager, Gitte Linderoth, Marianne Sjølin Frederiksen, Fredrik Folke, Annette Kjær Ersbøll, Hejdi Gamst-Jensen, Dina Cortes.

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
