## [Editor Report · Decision Letter 0]

26 Apr 2021

PONE-D-21-06235

Video triage of children with respiratory symptoms at a medical helpline is safe and feasible – a prospective quality improvement study

PLOS ONE

Dear Dr. Gren,

Thank you for submitting your manuscript to PLOS ONE. After careful consideration, we feel that it has merit but does not fully meet PLOS ONE’s publication criteria as it currently stands. Therefore, we invite you to submit a revised version of the manuscript that addresses the points raised during the review process.

The authors report interesting data regarding the use of video triage of children with respiratory illness; however, the information provided about parental satisfaction is better served by including it in the companion paper that was submitted.I suggest reworking this paper so that it focuses strictly on the patient outcomes. Also, please refrain from indicating that a measure "increased" or "decreased" when, in fact, there is no statistical difference. The wording implies that there was a change but if the measures do not achieve statistical significance, this is is not so.

We look forward to receiving your revised manuscript.

Kind regards,

Richard Bruce Mink

Academic Editor

PLOS ONE

Journal Requirements:

2. Please provide additional details regarding participant consent. In the ethics statement in the Methods and online submission information, please ensure that you have specified (i) whether consent was informed and (ii) what type you obtained (for instance, written or verbal, and if verbal, how it was documented and witnessed). If your study included minors, state whether you obtained consent from parents or guardians. If the need for consent was waived by the ethics committee, please include this information.

4. We note that Figure 2 includes images of participants in the study. 

As per the PLOS ONE policy (http://journals.plos.org/plosone/s/submission-guidelines#loc-human-subjects-research) on papers that include identifying, or potentially identifying, information, the individual(s) or parent(s)/guardian(s) must be informed of the terms of the PLOS open-access (CC-BY) license and provide specific permission for publication of these details under the terms of this license.

Please download the Consent Form for Publication in a PLOS Journal (http://journals.plos.org/plosone/s/file?id=8ce6/plos-consent-form-english.pdf).

The signed consent form should not be submitted with the manuscript, but should be securely filed in the individual's case notes.

Please amend the methods section and ethics statement of the manuscript to explicitly state that the patient/participant has provided consent for publication: “The individual in this manuscript has given written informed consent (as outlined in PLOS consent form) to publish these case details”.
---

## [Author Response · Author response to Decision Letter 0]

30 Jun 2021

Dear Richard Bruce Mink

On behalf of all the authors, I thank you for your interest in our manuscript ”Video triage of children with respiratory symptoms at a medical helpline is safe and feasible – a prospective quality improvement study” (PONE-D-21-06235).

We are today submitting a revised version, changed according to your points raised. Thus, we have moved all data regarding user satisfaction to the accompanying paper (“We can’t do without it”: parent and call-handler perceptions on video triage of children at a medical helpline, PONE-D-21-08760), and this present paper now solely focuses on patient outcome. We have also changed wordings such as decreased and increased when the statistical tests do not support it. The additional requirements, such as style requirements and consent forms for clinical pictures have also been fulfilled. We would like to underline the fact that, as mentioned in the manuscript, sharing of sensitive patient data originating from quality improvement data is prohibited in the Danish legislation, but an anonymized minimal data set has been prepared and can be requested by addressing the contact person mentioned in the manuscript.

We sincerely hope that this revised manuscript is accepted for publication, as we believe that it makes an important contribution to the knowledge of using video streaming in health services, a tool getting steadily increased focus worldwide.

Kind regards,

Dr. Caroline Gren

---

## [Decision Letter · Decision Letter 1]

9 May 2022

PONE-D-21-06235R1Video triage of children with respiratory symptoms at a medical helpline is safe and feasible – a prospective quality improvement studyPLOS ONE

Dear Dr. Gren,

Thank you for submitting your manuscript to PLOS ONE. After careful consideration, we feel that it has merit but does not fully meet PLOS ONE’s publication criteria as it currently stands. Therefore, we invite you to submit a revised version of the manuscript that addresses the points raised during the review process.

We look forward to receiving your revised manuscript.

Kind regards,

Wubet Alebachew Bayih, M.Sc.

Academic Editor

PLOS ONE

Journal Requirements:

Reviewers' comments:

Reviewer's Responses to Questions

**Comments to the Author**

1. If the authors have adequately addressed your comments raised in a previous round of review and you feel that this manuscript is now acceptable for publication, you may indicate that here to bypass the “Comments to the Author” section, enter your conflict of interest statement in the “Confidential to Editor” section, and submit your "Accept" recommendation.

Reviewer #1: (No Response)

2. Is the manuscript technically sound, and do the data support the conclusions?

Reviewer #1: Yes

3. Has the statistical analysis been performed appropriately and rigorously? 

Reviewer #1: Yes

4. Have the authors made all data underlying the findings in their manuscript fully available?

Reviewer #1: Yes

5. Is the manuscript presented in an intelligible fashion and written in standard English?

Reviewer #1: Yes

6. Review Comments to the Author

Reviewer #1: Thank you for the opportunity to review this manuscript. I was not a reviewer for the initial submission but have read both versions. This is a great study concept and design. Congratulations on this very important work. See specific minor edits below. Line number references are from the revised document with Track Changes.

One primary concern I have is with the use of the word “feasible” – in that nearly 10%, or 70 of the 734 calls included, either declined video triage or were unsuccessful in video streaming (data is from Figure 1). My concern is that those who declined may have done so due to lack of smartphone (line 132) or other device, or reliable internet to achieve this. Either way, it appears that video is not feasible for ~10% of the population sampled. This should be commented on in the discussion.

Additionally, for the ones triaged to video, can you please comment on how many were started as video and then could not be successfully completed?

This is partially addressed in Figure 1, but could be improved. I do recommend that you write about this relevant detail from this figure in the manuscript narrative, and not only refer to the Figure.

Please provide brief details about the intervention beyond “the technology is described in detail elsewhere” – it would serve the readers to have a brief summary, even if one sentence, for clarity in this manuscript.

Patient outcome (line 242): in this section, it is not clear to the reader whether within 8 hours the patients assessed in the hospital were because of the recommendation of the triaging call/video. Please be more clear about proportion of patients who were triaged to each outcome versus those who were triaged to one outcome but then presented to the hospital anyway - because of clinical condition or other; this needs to be fleshed out in the discussion as well, and would be beneficial in the paragraph starting at line 333

The results presented do not support that added assertion that video is “acceptable” (conclusion, line 432), given that parent and provider perspectives are removed from this paper.

Otherwise, I do recommend an additional review for grammar and syntax. For instance, make sure that sentences that start with numbers have them spelled out (e.g. line 235 “There 63% of … “ should read “Sixty-three percent of…” and delete the “There”). Another example is the parenthetical reference to a different paper that you have submitted (line 85 and line 163, “reference to accompanying paper”). There are other grammatical errors throughout the manuscript, please review and correct.

I very much look forward to your revisions and to seeing this published!

7. PLOS authors have the option to publish the peer review history of their article (what does this mean?). If published, this will include your full peer review and any attached files.

Reviewer #1: No

---

## [Author Response · Author response to Decision Letter 1]

18 Jun 2022

Please see the attached "response to reviwers" document, where all comments have been responded to.

---

## [Decision Letter · Decision Letter 2]

24 Nov 2022

PONE-D-21-06235R2Video triage of children with respiratory symptoms at a medical helpline is safe and feasible – a prospective quality improvement studyPLOS ONE

Dear Dr. Gren,

Thank you for submitting your manuscript to PLOS ONE. After careful consideration, we feel that it has merit but does not fully meet PLOS ONE’s publication criteria as it currently stands. Therefore, we invite you to submit a revised version of the manuscript that addresses the points raised during the review process.

ACADEMIC EDITOR:The authors should give informative description of how the measurement of their findings was done. I mean you should explain your methodology both in the abstract and methods section of the manuscript.

We look forward to receiving your revised manuscript.

Kind regards,

Wubet Alebachew Bayih, M.Sc.

Academic Editor

PLOS ONE

Journal Requirements:

Reviewers' comments:

Reviewer's Responses to Questions

**Comments to the Author**

1. If the authors have adequately addressed your comments raised in a previous round of review and you feel that this manuscript is now acceptable for publication, you may indicate that here to bypass the “Comments to the Author” section, enter your conflict of interest statement in the “Confidential to Editor” section, and submit your "Accept" recommendation.

Reviewer #2: All comments have been addressed

2. Is the manuscript technically sound, and do the data support the conclusions?

Reviewer #2: Yes

3. Has the statistical analysis been performed appropriately and rigorously? 

Reviewer #2: (No Response)

4. Have the authors made all data underlying the findings in their manuscript fully available?

Reviewer #2: Yes

5. Is the manuscript presented in an intelligible fashion and written in standard English?

Reviewer #2: (No Response)

6. Review Comments to the Author

Reviewer #2: Thank you for inviting me to review this manuscript entitled on: Video triage of children with respiratory symptoms at a medical helpline is safe and feasible – a prospective quality improvement study

The study aimed at to study the safety and feasibility of introducing video triage of young children with respiratory symptoms at a medical call center, as well as the impact on patient outcome. The authors have raised very important issue. After the authors address the following concerns, it will be suitable for publication to Plose one journal

Questions and comments

The abstract is somehow complex to understand. It would be better if authors make it clear. It lacks some points which must be included in the method section of the abstract. It doesn’t sate clearly about the data collection tools, study period, final sample size, study design and model of data analysis

Why children’s less than six months are excluded from the study?

Explain how the limitation of this study affect your result.

How the authors did model testing? if no justify, if yes state clearly.

How is the validity of the data collection tool? was it validated? if yes reference, if no justify

7. PLOS authors have the option to publish the peer review history of their article (what does this mean?). If published, this will include your full peer review and any attached files.

Reviewer #2: No

---

## [Author Response · Author response to Decision Letter 2]

19 Dec 2022

-Thank you for inviting me to review this manuscript entitled on: Video triage of children with respiratory symptoms at a medical helpline is safe and feasible – a prospective quality improvement study.

The study aimed at to study the safety and feasibility of introducing video triage of young children with respiratory symptoms at a medical call center, as well as the impact on patient outcome. The authors have raised very important issue. After the authors address the following concerns, it will be suitable for publication to Plose one journal

-We thank you for your time and effort in reviewing our manuscript, and we hope that you will find our comments and corrections satisfying. We too find this an important issue that deserves attention.

-The abstract is somehow complex to understand. It would be better if authors make it clear. It lacks some points which must be included in the method section of the abstract. It doesn’t sate clearly about the data collection tools, study period, final sample size, study design and model of data analysis

-We have now added the wanted information (data collection tools, study period, sample size and data analysis) to the abstract.

Final sample size: 617; study design: prospective quality improvement study with patients enrolled to video or standard telephone triage (1:1); data analysis: “Logistic regression was used to test the effect on outcomes.” (this is the main method used); and finally, study period (February 2019-March 2020) has been added to abstract and results section, thank you for pointing this out.

-Why children’s less than six months are excluded from the study?

-Yes, this is a relevant question, thank you. We chose 6 months as a cut-off due to infants relatively more often having severe illness, and we therefore did not want to take any chances in this early phase, before we were sure that video triage was safe. Also, infants are inherently harder to assess than older children, and are generally recommended to be assessed face-to-face. We have added a section about this at lines 123-126.

-Explain how the limitation of this study affect your result.

-Yes, this is a very relevant discussion. The most important limitation, that the study was not conducted as a randomized controlled study, due to the call-handlers not having time to inform the patients sufficiently for a written informed consent, and difficulties of setting the computers up to do the randomization, conveys the possibility that the two groups (video and telephone triage groups) are not completely alike. But as age, gender, symptoms and hospital outcome are similar, we reckon that the groups are comparable. But of course, a randomized controlled trial would have been preferable. 

Another important limitation is that we did not reach the calculated sample size, due to the Corona pandemic. Had we reached the sample size calculated beforehand, we believe that we would have reached a significant result concerning how many patients that were assessed at hospital after the call. Lastly, we cannot know how video triage works in patients in other ages or with other symptoms, this must be tested in other studies, and that would be interesting to investigate. However, we tested video triage in the most frequently occurring pediatric population at this medical helpline, i.e. young children with symptoms from the airways, so the results should thus be representative for a large part of pediatric calls. 

This discussion can be found in the limitations-section as well. 

-How the authors did model testing? if no justify, if yes state clearly

-We tested the model during the first two weeks of the study, the pilot phase, where we evaluated how the nurses experienced the set-up and how the technical solution worked. Afterwards we made some smaller adjustments. 

Also, before this study was launched, a study at the Danish emergency helpline was using the same technical set-up (GoodSAM instant-on-scene) and we therefore could use some the experiences gained from that study. 

-How is the validity of the data collection tool? was it validated? if yes reference, if no justify

-We are not completely certain what is understood by “data collection tool” in this context. However, the surveys answered by the call-handlers were not validated more than by oral feedback from the call-handlers during the pilot period regarding for example clarity of wording. 

Concerning the internal electronic patient records at MH1813, they are used every day and maintained by a data management department, and data concerning all the participating children was found and was complete. 

Data collection from the hospitals’ electronic charts were done manually by the first author, and all children were identified, and all wanted data was able to be obtained. Generally, Danish registers are renowned to be complete and reliable.

---

## [Decision Letter · Decision Letter 3]

4 Apr 2023

Video triage of children with respiratory symptoms at a medical helpline is safe and feasible – a prospective quality improvement study

PONE-D-21-06235R3

Dear Dr. Gren,

We’re pleased to inform you that your manuscript has been judged scientifically suitable for publication and will be formally accepted for publication once it meets all outstanding technical requirements.

Kind regards,

Wubet Alebachew Bayih, M.Sc.

Academic Editor

PLOS ONE

Additional Editor Comments (optional):

Reviewers' comments:

Reviewer's Responses to Questions

**Comments to the Author**

1. If the authors have adequately addressed your comments raised in a previous round of review and you feel that this manuscript is now acceptable for publication, you may indicate that here to bypass the “Comments to the Author” section, enter your conflict of interest statement in the “Confidential to Editor” section, and submit your "Accept" recommendation.

Reviewer #2: All comments have been addressed

2. Is the manuscript technically sound, and do the data support the conclusions?

Reviewer #2: Yes

3. Has the statistical analysis been performed appropriately and rigorously? 

Reviewer #2: Yes

4. Have the authors made all data underlying the findings in their manuscript fully available?

Reviewer #2: Yes

5. Is the manuscript presented in an intelligible fashion and written in standard English?

Reviewer #2: Yes

6. Review Comments to the Author

Reviewer #2: All my concerns were fully addressed . Thank you very much for your commitment and motivation . You have raised an important issue .

7. PLOS authors have the option to publish the peer review history of their article (what does this mean?). If published, this will include your full peer review and any attached files.

Reviewer #2: No

---

## [Editor Report · Acceptance letter]

10 Apr 2023

PONE-D-21-06235R3 

Video triage of children with respiratory symptoms at a medical helpline is safe and feasible – a prospective quality improvement study 

Dear Dr. Gren:

I'm pleased to inform you that your manuscript has been deemed suitable for publication in PLOS ONE. Congratulations! Your manuscript is now with our production department. 

Kind regards, 

on behalf of

Dr. Wubet Alebachew Bayih 

Academic Editor

PLOS ONE